# Mediation of Osseointegration, Osteoimmunology, and Osteoimmunologic Integration by Tregs and Macrophages: A Narrative Review

**DOI:** 10.3390/ijms26115421

**Published:** 2025-06-05

**Authors:** Jong Il Yun, Su In Yun, Jae Hong Kim, Duk Gyu Kim, Deok-Won Lee

**Affiliations:** 1Dental R&D Center, Zerone Cellvane Inc., Seoul 04363, Republic of Korea; oralyun@naver.com (J.I.Y.); flavan@naver.com (J.H.K.); 2Department of Chemistry, University of Pittsburgh, Pittsburgh, PA 15213, USA; raceu0422@gmail.com; 3Zerone Cellvane Inc., Cheonan 31116, Republic of Korea; abramsax@naver.com; 4TheOneOMS, Seoul 04363, Republic of Korea

**Keywords:** T-regulatory cell, adenosine, macrophage, osseointegration, osteoimmunology, osteoimmunologic integration

## Abstract

Osseointegration is the direct contact between living bone and a dental implant, with supporting evidence confirming the direct connection between bone and titanium, found using an electron microscope. However, the fundamental mechanisms and interconnections between the bone and titanium are not clearly understood. At present, osteoimmunology explores the interaction between bone and immune cells not only in the medical field but also in dentistry. Immunology in bone cell formation has long been a research topic; however, interest in these effects has recently surged. Through subsequent studies, osteoimmune reaction occurs in response to dental implant insertion into the bone and this mechanism portrays more accurate tissue response compared to the traditional term osseointegration. Additionally, osseointegration is a foreign body defense mechanism to protect the implant when bone forms at the contact surface between the dental implant and the alveolar bone. The term “osteoimmunology” refers to the relationship between the immune system and bone tissues. Understanding osteoimmunologic concepts may enable the development of immunomodulatory strategies to improve, maintain, and ultimately restore osseointegration. In order for biocompatible materials such as dental implants to settle and be maintained in the body, it is necessary to understand the complex interrelationships of the bone immune environment, which will enable the development of biomaterials that are more favorable to osteoimmune environments. Therefore, this review presents previous insights into cellular and molecular interactions between bone and the immune system, specifies the roles of T-regulatory cells (Tregs) and macrophages, and demonstrates their potential for translational applications worldwide.

## 1. Introduction

### Osteoimmunology—Fundamentals of Immunity and Immune Cells

Immunity is a natural defense mechanism that immediately protects the body by distinguishing between self and non-self cells and eliminates the non-self cells. This biological defense system includes immune cells that respond to external pathogens such as viruses and bacteria, as well as to the body’s self components, which typically leads to autoimmune disease. Therefore, it is critical to understand that proper immune regulation must encompass both defense mechanisms against external pathogens and the modulation of responses to the body’s self components. The human body strictly recognizes self and non-self components and maintains its establishment through the elimination of external particles. The body tries to remove antigens and return to its original state by proliferating immune cells that specifically respond to antigens. As an example, bacteria that are recognized as non-self are often removed. In addition, the immune system continues to monitor itself and tolerate such cells as part of the body. However, when cells undergo slight variations in their components, they are immediately targeted for apoptosis. Ultimately, the immune system operates invariably to maintain a body’s stable internal state, also known as homeostasis [1,2,3,4,5].

The human body contains two defense mechanisms against various pathogens: innate immunity and adaptive immunity [6]. Innate immunity serves as the first line of defense which is activated immediately upon pathogen invasion and is also referred to as natural immunity or primary immunity. This immune system is composed of cells such as neutrophils, macrophages, dendritic cells, eosinophils, basophils, mast cells, and natural killer (NK) cells. When the primary immune system is penetrated, the second line of defense is activated. This is known as adaptive or acquired immunity, and can also be referred to as secondary immunity. This system is primarily represented by T cells and B cells, and produces a definite and long-lasting immune response [7].

The immune cells responsible for immunity originate from hematopoietic stem cells (HSCs) which are stem cells that generate blood cells. This process is known as hematopoiesis, which primarily occurs in the red bone marrow, located in the spongy, cancellous tissue within the bone, as shown in Figure 1. HSCs produce various types of blood cells through two major lineages: the myeloid lineage and the lymphoid lineage. The myeloid lineage includes monocytes, macrophages, dendritic cells, neutrophils, eosinophils, basophils, mast cells, erythrocytes (red blood cells), and megakaryocytes, which are a precursor to platelets. The lymphoid lineage is composed of T cells, B cells, and natural killer (NK) cells. Among these, T cells play a central role in adaptive immunity. There exist various types of T cells, such as cytotoxic T lymphocytes (CTLs), CD8+ T cells, and killer T cells, which target tumor and infected cells. There are also T helper cells—also known as Th cells or CD4+ Tcells—that act as a commander by attracting innate immune cells and regulate immune response by coordinating both innate and adaptive immune cells [8].

## 2. T-Regulatory Cell (Tregs) Interactions with Other Immune Cells

### 2.1. T-Helper17 (Th17)/Treg

Th cells are mature T-cells that trigger the surface protein cluster of differentiation 4 (CD4) and release cytokines to regulate other immune cell activities. These are often classified into various subtypes such as T-helper 1 (Th1), Th2, Th17, and Treg cells [9]. Previously, Th1 cytokines were associated with inflammatory bone destruction while Th2 cytokines, also known as classical antagonists, were thought to minimize bone resorption [10,11]. However, subsequent studies have questioned the roles of Th1 and Th2, and have proven the presence of Th1 and Th2 cytokines in human periodontitis lesions [12,13]. Therefore, the cause of the pathogenesis of periodontitis cannot be fully supported through the imbalance of Th1/Th2. More recently, the significance of Th17 and Tregs has been highlighted. Th17 are cytokine-producing cells associated with inflammatory and numerous autoimmune diseases [14]. They induce the expression of receptor activator of NF-κB ligand (RANKL) in osteoblasts and fibroblasts, release the key cytokine interleukin-17 (IL-17) to enhance local inflammation, and increase the production of inflammatory cytokines such as IL-21, tumor necrosis factor-α (TNF-α), IL-1β, and IL-6, which triggers faster RANKL expression. As a result, Th17 are the major contributors of bone loss in periodontitis and peri-implantitis [15,16].

Tregs play an extensive role in stimulation and suppression of immune cells to maintain the cell homeostasis [17]. Tregs inhibit the activation of T conventional (Tconv) by elevating the expression of CD25, the α chain of the IL-2 receptor, and binding it to IL-2 [18]. To date, several categories of Tregs have been identified, with the two main types being thymus-derived forkhead box P3+ (Foxp3+) Tregs and inducible Tregs [19]. However, the over-expression of Th17 is a major cause of periodontitis and peri-implantitis, accompanied by alveolar bone resorption. To minimize this effect, the anti-inflammatory functions of Tregs can be utilized [20].

In cases of periodontitis and peri-implantitis, appropriate regulation of immune response prevents the spread of pathogenic microorganisms and simultaneously avoids additional collateral tissue damage. Therefore, Tregs are primarily drawn to the infected tissue to control the immune response [21,22]. Tregs regulate bone metabolism by secreting transforming growth factor-β (TGF-β), interleukin-10 (IL-10), IL-4, and cytotoxic T lymphocyte-associated antigen-4 (CTLA-4) and directly inhibiting osteoclast production. When peri-implantitis progresses, the proportion of Th17 increases. It is known that in the early stages, Tregs maintain a balance with Th17, but in the later stages, as the proportion of Tregs decreases, there is an immune imbalance, leading to bone resorption. Hence, Foxp3+Tregs play a crucial role in regulating periodontitis and peri-implantitis [23,24].

### 2.2. Tregs and Adenosine

Adenosine is a stable purine nucleoside that is typically found at low levels during regular physiological conditions, with a biological half-life lasting less than 10 s [25]. Under metabolic stress conditions, the adenosine levels in plasma increase. Blood adenosine triphosphate (ATP) is dephosphorylated to adenosine by ectonucleoside triphosphate diphosphohydrolase-1 (CD39) and ecto-5′-nucleotidase (CD73). Equilibrative nucleoside transporter 1 (ENT1) releases intracellular adenosine to the extracellular space [26]. It is commonly understood that adenosine or its analogs block inflammation in various organs such as the liver, lungs, kidneys, heart, and gastrointestinal tracts [27]. In the extracellular environment, adenosine binds to a type of G protein-coupled receptor (GPCR) known as adenosine receptors, which are subdivided into four different categories: A1, A2A, A2B, and A3. Here, A1 and A2A adenosine receptors (A2AAR) are high-affinity receptors, while A2B and A3 adenosine receptors (A3AR) are low-affinity receptors. All of these receptors are regulated through adenylyl cyclase (AC) activity, which controls intracellular cyclic adenosine monophosphate (cAMP) levels. A2AAR and A2BAR stimulate AC, whereas A1AR and A3AR inhibit AC [28]. A2AAR is commonly noticed in various tissue cells, but it is mainly identified in leukocytes, platelets, and vascular tissues. It mediates anti-inflammatory responses and proliferates vasodilation [29]. The binding of adenosine and A2AAR is the most rapid binding mechanism to protect tissues from external damages and dramatically reduces the production of inflammatory cytokines [30]. Additionally, A2BAR is widely expressed peripherally and found in various cells which regulate inflammatory and immune response under pathological conditions [29].

Tregs can produce massive amounts of extracellular adenosine via coupling the A2A adenosine receptor (A2AAR) while inhibiting the T effector cells. They express CD39 and CD73, which enables the production of adenosine from ATP. Adenosine production then controls Th17 responses [31]. Moreover, adenosine can optimize the immunomodulatory activity of Tregs through A2AAR. The increased activity of A2AAR then significantly enhances the expression of co-inhibitory molecule programmed cell death protein-1 (PD-1) [32]. These immune regulations caused by A2AAR are mediated by an increase in intracellular cyclic adenosine monophosphate (cAMP) levels [33]. Therefore, under inflammatory conditions, Tregs increase the production of adenosine, which in return enhances the immunosuppressive activity of Tregs. Ultimately, the resulting feedforward mechanism successfully suppresses immune responses [34].

### 2.3. Tregs and Neutrophils

Neutrophils are the frontline soldiers in the injury sites which are actively recruited in the healing process. However, when there is an overabundance of neutrophils, this could possibly harm unwanted cells. Along with monocytes (immature cells) and macrophages (mature cells), neutrophils remove necrotic tissues and cellular debris, and release cytokines to regulate chemotaxis for acute inflammation. Not only do they contribute to various chronic inflammatory diseases, but neutrophils also exacerbate these conditions through the release of proteases and the formation of neutrophil extracellular traps (NETs) [35,36,37,38]. Here, Tregs can reduce the neutrophils and modulate T helper cells to indirectly regulate the inflammation and tissue regeneration. To do so, Tregs induce anti-inflammatory molecules such as IL-10, TGF-β, heme oxygenase-1 (HO-1), and indoleamine 2,3-dioxygenase (IDO) secreted inside the neutrophils [39,40]. Also, Tregs promote the apoptosis of neutrophils and adjust their infiltration [41,42]. Tregs affect neutrophil activities, inflammation, and tissue healing.

### 2.4. Tregs and Dendritic Cells (DCs)

Dendritic cells (DCs) are antigen-presenting cells that capture oral microorganisms, migrate to the lymph nodes, and regulate T helper cell (CD4+) differentiation; they are thus essential for the progression of periodontitis and peri-implantitis [43]. The inhibitory receptor CTLA-4 on regulatory Tregs binds to co-stimulatory molecules (CD80 and CD86) on dendritic cells, which inhibits their antigen-presenting function and induces IDO generation. This escalates T-cell apoptosis [44]. When Tregs increase adenosine, they act on A2A adenosine receptors (A2AAR) to inhibit pro-inflammatory activation and on A2B adenosine receptors (A2BAR) to induce tolerogenic conditions in antigen-presenting cells like DC [45]. Tregs also secrete IL-10 and TGF-β to inhibit the expression of HLA-DR (Human Leukocyte Antigen—DR isotype), CD80/86, and CD40 on the DC surface and the production of inflammatory cytokines such as TNF-α and IL-12 [46]. IL-35 induces tolerogenic DCs by increasing the expression of CD11b and IL-10 and decreasing major histocompatibility complex-II (MHC-II) [47]. Tregs can make additional contact through extracellular vesicles (EVs) that contain miR-150-5p and miR-142-3p to mediate the non-contact regulation of DCs. This would lead to an increase in IL-10 and a decrease in IL-6 production in DCs, which promotes a tolerogenic DC state [48].

### 2.5. Tregs and Mesenchymal Stem Cells (MSCs)

MSCs, progenitor cells of many tissue cells such as osteoblasts and adipocytes, have immunosuppressive mechanisms very similar to Tregs and interact with them [49]. MSCs heavily depend on the Jagged-1 gene to increase Foxp3+Tregs or convert Foxp3-Tconv to Foxp3+Tregs [50]. Tregs and MSCs cooperate and follow the CD39-CD73-adenosine production pathway to increase adenosine [51,52]. Undifferentiated bone marrow-derived MSCs primarily express A2BAR, which is vital for differentiating MSCs into osteoblasts (OBs). The presence of adenosine reacts to A2BAR derived from MSCs and enhances OB classification [53]. It can also excite A2AAR derived from OBs to increase the secretion of IDO and HO-1 for immunosuppression and tissue regeneration [54]. Ultimately, Tregs regulate RANKL and osteoprotegerin (OPG) production from OBs to further influence osteoclast formation. Therefore, Tregs’ role is crucial in identifying the precise RANKL/OPG ratio that signals OB differentiation and bone formation [55].

### 2.6. Tregs and Macrophages

Tregs can interact with other major innate immune cells such as macrophages for inflammatory responses. Macrophages have a key role in autophagy, which removes apoptotic neutrophils and other cells. Additionally, they contribute to tissue repair and regeneration [56]. Monocyte-derived macrophages exist in two subsets: M1 and M2. M1 are pro-inflammatory macrophages induced by interferon-γ (IFN-γ) or TNF-α, while M2 are anti-inflammatory macrophages induced by IL-4/IL-13 or IL-10. Here, Tregs are an important regulator for expressing the macrophage phenotypes. They alleviate tissue damage, regulate macrophage activity, and escalate tissue survival to maintain tissue repair and homeostasis [57]. Tregs utilize various types of cytokine secretion or cell interactions to induce monocytes’ phagocytic capacity and markers such as CD206, CD163, and HO-1 while differentiating monocytes into the anti-inflammatory M2 phenotype. This reveals the low expression of CD40, CD80/86, and Class II MHC. Tregs can also increase the secretion of anti-inflammatory cytokines such as IL-4, IL-10, IL-13, and TGF-β and suppress the secretion of pro-inflammatory cytokines like TNF-α, nitrogen oxide (NO), and reactive oxygen species (ROS) [58,59]. In the Treg-mediated immune suppression, there exist various cell-contact regulators. For example, CTLA-4 interacts with co-stimulatory receptors CD80/86, PD-1 binds to PD-L1, lymphocyte antigen gene-3 (LAG-3) binds to MHC-II, T cell immunoreceptor with Ig and ITIM domains (TIGIT) binds with CD155/122, FasL induces Fas+ cell apoptosis, and neuropilin 1 (Nrp1) monitors Tregs’ Foxp3 activation and deactivation [60]. However, for the Treg-mediated macrophage inflammation suppression to occur, a signaling pathway via Krüpel-like factor 10 (KLF10) and mammalian target of rapamycin complex 1 (mTORC1) is required [61,62]. Tregs are responsible for directly producing EVs on the cell surface, which leads to the emergence of CD73 in EVs. This triggers the conversion of the extracellular AMP into adenosine and regulates immune responses by inhibiting M1 macrophages and promoting M2 macrophages [63] (Figure 2).

## 3. M1/M2 Polarization in Dental Implants

Macrophages are monocyte-derived mature immune cells that exhibit high plasticity. Macrophage polarization is the process in which macrophages acquire distinct functional characteristics in response to specific stimuli from the extracellular microenvironment. This typically results in two phenotypic changes: classically activated M1 (pro-inflammatory) macrophages and alternatively activated M2 (anti-inflammatory) macrophages [64]. At present, the function and phenotypic polarization of macrophages is a key research area to understand inflammatory conditions that lead to diseases like atherosclerosis, type 2 diabetes, obesity, and periodontitis. Due to their phagocytic activity and high cellular plasticity, macrophages are necessary for establishing homeostasis and identifying diseases. When there are various environmental signals, macrophages regulate different phenotypes to incite or alleviate the inflammation. For an example, when macrophages are activated by lipopolysaccharide (LPS) or IFN-γ, they exhibit the M1 phenotype which is associated with inflammatory responses, phagocytosis, tissue destruction, and the production of IL-6 and IL-1β. Alternatively activated macrophages with the M2 phenotype are associated with anti-inflammatory responses, which include the production of IL-10 and TGF-β, inducing tissue repair and angiogenesis [65].

Numerous studies have revealed that the increased expression of inflammation by M1 macrophages is associated with the pathogenesis of periodontal and peri-implant diseases. Periodontitis and peri-implantitis are related to the changes in M1 and M2 phenotypes, where the phenotype shift from M2 to M1 causes critical damage to the periodontal tissue, including the alveolar bone [66,67]. M1 macrophages are responsible for secreting inflammatory mediators such as TNF to increase the number of osteoclast precursors and promote their differentiation to directly regulate osteoclastogenesis. M1 macrophages can indirectly up-regulate RANKL production through osteoblasts and other stromal cells, which stimulate osteoclast formation and activation via the RANKL/OPG/RANK signaling pathway [68]. Meanwhile, local actions of IL-4 and IL-13 increase M2 macrophages and augment bone volume to enhance bone healing effects [69]. M2 macrophages can produce the potent anti-inflammatory cytokine IL-10, which inhibits early osteoclast development and prevents continuous pre-osteoclast (preOC) development [70]. If M1 polarization is prolonged, it can lead to increased release patterns of M2 macrophage fibrotic-promoting cytokines while resulting in the formation of fibrous capsules [71]. Therefore, for optimal bone remodeling during bone healing, the inflammatory microenvironment must be timely adjusted to an anti-inflammatory stage and M1 macrophages need to be converted to M2 macrophages around the implant site [72].

Osteoimmunology in bone metabolism has recently gained attention in the field of dentistry, and the osseointegration of implants is understood as a state of foreign body equilibrium (FBE). This process heavily depends on the complex cellular heterogeneity and dynamic changes in the implant-mediated osteoimmune microenvironment. Therefore, the balanced plasticity of macrophages surrounding the implant, known as innate immune cells, is related to long-term FBE. On the other hand, the increased proportion of M1/M2 near the implant will affect bone resorption and is likely to indicate an early or ongoing clinical manifestation of foreign body reaction (FBR) [73].

In osteoimmunology, there is an increasing trend in the development of drugs and advances in implant surfaces utilizing M1/M2 polarization [74,75]. Polydeoxyribonucleotide (PDRN), a DNA fragment extracted from salmon sperm and testes, is a drug that responds to adenosine receptors on the cell surfaces to regulate inflammation and escalate tissue regeneration [76]. PDRN promotes the secretion of IL-10 and VEGF derived from M2 macrophages and converts M1 macrophages into M2 macrophages phenotypically [77]. In a study using bone graft composites containing PDRN, the inclusion of PDRN attenuated osteoclast differentiation induced by RANKL [78]. Moreover, when M2 macrophage polarization was increased, this activated angiogenesis and tissue remodeling, which further signaled the secretion of cytokines, chemokines, and growth factors. Simultaneously, M1 polarization was stopped to reduce inflammatory cytokines, which indicated improved tissue regeneration in the inflammatory environment [79] (Figure 3).

## 4. Immunologic Understanding of Osseointegration (Osteoimmunologic Integration: OII)

In 1981, T. Albrektsson defined osseointegration as the direct contact between living bone and an implant, with supporting evidence confirming the direct connection between bone and titanium found using an electron microscope. However, the fundamental mechanisms and interconnections between the bone and titanium were not clearly understood [80]. Following his subsequent studies, Albrektsson stated that osteoimmune reaction occurs in response to implant insertion into the bone and this mechanism portrays more accurate tissue response compared to the traditional term osseointegration. Additionally, Albrektsson described osseointegration as a foreign body defense mechanism to protect the implant when bone forms at the contact surface between the implant and the alveolar bone [73].

Clinical and animal studies indicate that many immune cell types, including T cells, macrophages, and neutrophils, are involved in fracture healing by coordinating a chain of events that regulate bone formation and remodeling. Immune cells, including T cells, are involved in fracture healing, regulating the microenvironment and preparing MSCs in the inflammatory phase, and coordinating bone formation and resorption in the bone remodeling phase [81].

In 2023, Waad Kheder showed that the infiltration of immune cells in the tissues surrounding dental implants plays an important role. There is an increase in lymphocyte and macrophage infiltration in the tissues around the failed implant. In particular, it is associated with T-cell-mediated immunity and M1/M2 macrophage polarization. These results show evidence that titanium particles regulate the polarization of lymphocytes and macrophages in the gingival tissue around the implant, which may help to understand the imbalance in osteoblast and osteoclast activity and the failure of osseointegration of dental implants [82].

The success of the osseointegration of dental implants directly into the alveolar bone in vivo greatly depends on the local immune microenvironment controlled by various immune cells and appropriate osteoimmune regulation [83]. New bone formation after dental implants typically occurs in three stages: the initial inflammation stage, the bone formation stage, and the remodeling stage. Here, macrophages are involved in all three stages to participate in bone healing and formation. More specifically, macrophages that reside in the bone endosteum participate in bone recovery [84,85]. When the implant is inserted into the soft and hard tissues, the surface absorbs blood proteins and triggers an inflammatory cascade operated by the innate immune system [86]. If the inflammation is not properly healed or reactivated, the implant is most likely going to fail [87]. According to recent studies, immature monocytes are known to directly convert into mature macrophages after biomaterial is implanted [88]. Circulating monocytes can infiltrate the inflammatory tissue and act as precursors of wound-healing macrophages. When monocytes enter the implantation site, cytokines and chemokines like IL-1, IL-8, monocyte chemoattractant protein-1 (MCP-1), CXCL13, and macrophage inflammatory protein (MIP) are released and stimulate further monocyte infiltration and macrophage activation [89]. During the early stage of development, the majority of macrophages express the M1 phenotype, but they are gradually converted to the M2 phenotype after interacting with multiple cells, cytokines, and extracellular matrix components. This increases osteoblast activity, which further leads to new bone tissue formation [90]. It takes approximately a month after surgery for osteoclasts to absorb the remnants of the bone tissue and for mesenchymal stem cells (MSCs) and osteoblasts to migrate to the bone formation area near the implant surface and begin mineralization. After 3 months post-implantation, the implant surface is surrounded by osteoblasts and osteocytes, which gradually mature and lead to progressive osseointegration [91].

In conclusion, osseointegration—which denotes the well-known combination of dental implant and alveolar bone—is the result of a recently identified osteoimmunologic response and, so, it is reasonable to understand osseointegration as “osteoimmunologic integration”. This may offer more valuable insights to potentially expand the field of dentistry (Figure 4).

## 5. Conclusions

In recent years, osteoimmunology has become fundamental to understanding the complex immunologic interactions among bone and implants, bone regeneration, and various diseases. Therefore, the interpretation of osseointegration is understood as an osteoimmune response. From this point of view, the immune system separates biomaterials (e.g., implants) from the living body, which effectively protects them from the bone tissue. This osseointegration is a continuous and dynamic biological reaction and is a concept that achieves equilibrium with foreign substances [73]. While several immune cells are involved in this process, macrophages play an important role in osseointegration dynamics. Macrophages derived from circulating monocytes and resident macrophages contribute to achieving an environment for promoting bone regeneration around the implant in the early stages of implant placement [92]. At this time, phenotypic transition from M1 macrophages to M2 macrophages appears to be very important in regulating bone formation. In addition, the efficient and timely transition from the M1 macrophage phenotype to the M2 macrophage phenotype releases osteogenesis cytokines and enables the formation of bone tissue around the implanted implant, which is the basis of the concept of osteoimmunologic regulation. Therefore, biomaterials with immunomodulatory abilities can create an immune environment that enhances osteogenesis and regulates proper osteoclast formation during bone remodeling processes. These properties are important for the development of immunomodulatory biomaterials. Biomaterials that lack immunomodulatory capacity can cause excessive inflammation and lead to an imbalance in osteoclast formation with respect to bone formation [93].

Understanding osteoimmunologic concepts may enable the development of immunomodulatory strategies to improve, maintain, and ultimately restore osseointegration. A number of studies have recently emerged based on this concept [75,90]. Topics included altering implant surface properties to convert macrophage phenotypes around implants from M1 to M2, reducing the secretion of inflammatory cytokines using implants ionized with LiCl or Mg, modulating macrophage phenotypes using polarizing cytokines such as IL-4, and promoting the innate immunomodulatory capacity of MSCs by mechanistic stimulation to the outside of the tissues surrounding the implant [92]. Recent in vivo studies show biodegradable Mg implants elicit a pronounced, transient M1 surge before any M2 shift [94,95].

In addition, studies that have explored complex material–cell interactions using machine learning to understand immune modulation by macrophage polarization have been presented [75]. PDRN is known as a DNA drug that regulates excess inflammation and aids tissue regeneration [76]. It has also been reported that PDRN polarizes M1 to M2 in the inflammatory environment [77]. Although many studies are still needed to support the usage of PDRN in the field of dentistry, it is considered a strong candidate drug that may be helpful in the early and ongoing maintenance of implant osseointegration in the future.

At present, it is accepted that an understanding of osteoimmunology is essential to understand bone metabolism accurately. In order for biocompatible materials such as implants to settle and be maintained in the body, it is necessary to understand the complex interrelationships of the bone immune environment, which will enable the development of biomaterials that are more favorable to osteoimmune environments.

## Figures and Tables

**Figure 1 ijms-26-05421-f001:**
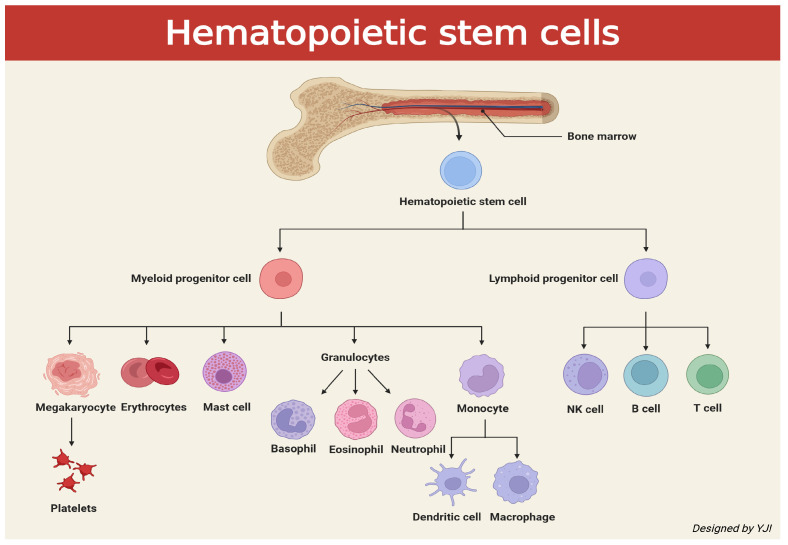
Schematic illustration depicting signaling cascades of cells caused by hematopoietic stem cells released from bone marrow (created in BioRender. Jong Il Yun. (2025) https://BioRender.com).

**Figure 2 ijms-26-05421-f002:**
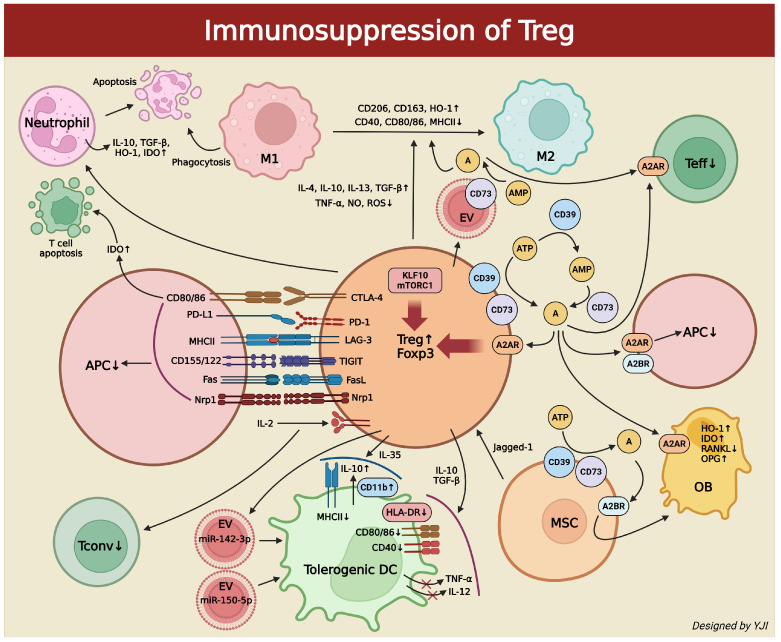
Schematic illustration depicting immunosuppressive methods of Tregs on immune cells (created in BioRender. Jong Il Yun. (2025) https://BioRender.com). ↑, increase; ↓,decrease; red X represents inhibition.

**Figure 3 ijms-26-05421-f003:**
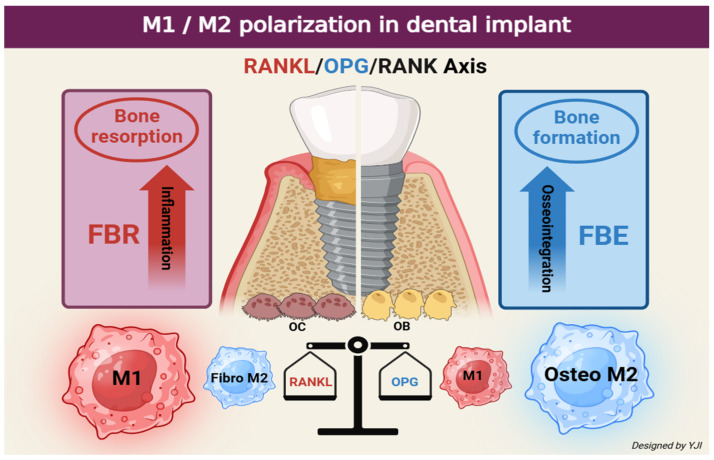
Schematic illustration depicting differences in bone response around the implant according to the phenotype change in M1 and M2 (created in BioRender. Jong Il Yun. (2025) https://BioRender.com).

**Figure 4 ijms-26-05421-f004:**
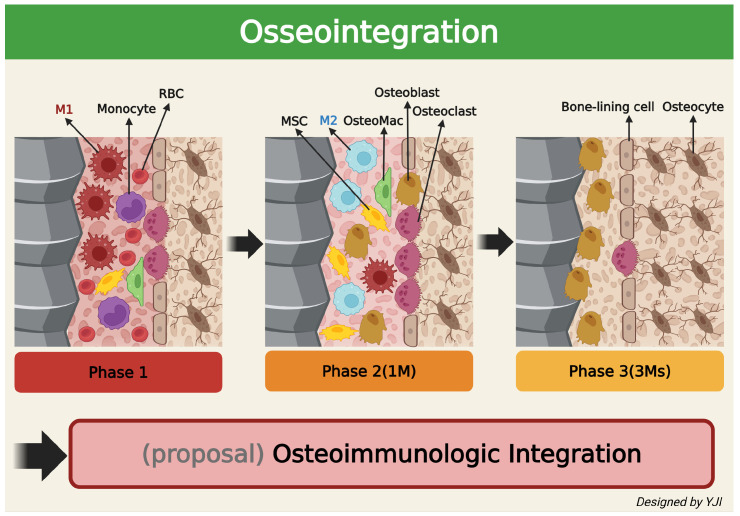
Schematic illustration depicting changes in the surrounding bone’s immune system over time after implantation in the living bone (created in BioRender. Jong Il Yun. (2025) https://BioRender.com).

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
