# Peer review of "Mediation of Osseointegration, Osteoimmunology, and Osteoimmunologic Integration by Tregs and Macrophages: A Narrative Review"

_ijms, 2025, doi:10.3390/ijms26115421_

Round 1

Reviewer 1 Report

Comments and Suggestions for Authors

This review synthesises current knowledge on T‑regulatory cells and macrophage phenotypes in osteoimmunology and osseointegration. The topic is timely, but several major limitations must be addressed:

  • The manuscript neither states a clear objective nor identifies a specific gap in the literature. Please begin the Introduction with a concise aim statement and explain what is unknown and why this review is needed now.
  • The opening paragraph mentions “oste­o­immunology in the field of dentistry,” yet no dental‑specific link is developed. Please make this connection explicit throughout or omit the title of this subsection in the Introduction.
  • General immunology of T cells/T regs is reviewed in the first half, but they disappear in the osseointegration section, which shifts to macrophages. Explain why T regs are not discussed in the peri‑implant context and why macrophages are not introduced earlier.
  • The literature covering immune response and osseointegration is selective and incomplete. The discussion relies almost exclusively on the “foreign‑body response” concept introduced by Albrektsson and colleagues, but provides no evidence or contrasting viewpoints. A large body of work on innate and adaptive immunology during metallic implant osseointegration (both dental and orthopaedic) is omitted. Please broaden the literature survey to include experimental and clinical studies that investigate macrophage and T‑cell dynamics beyond the Albrektsson paradigm; also please integrate findings from orthopaedic models where immune–bone crosstalk is well characterised
  • The title mentions T regs but not macrophages, although macrophages dominate the second half of the review. Please revise the title.
  • The abstract does not state explicit review questions or summarise how the manuscript answers them.  Please rewrite to state the aim/gap, to present the main thematic findings, and to conclude with specific knowledge gaps and clinical implications.
  • The review claims that magnesium promotes M2 macrophage polarisation. Recent in‑vivo studies show biodegradable Mg implants elicit a pronounced, transient M1 surge before any M2 shift. Please correct.

Author Response

Thank you for your honorable comment. It is good for us to make it better manuscrip.

Comments and Suggestions for Authors

This review synthesises current knowledge on T‑regulatory cells and macrophage phenotypes in osteoimmunology and osseointegration. The topic is timely, but several major limitations must be addressed:

Comments 1: The manuscript neither states a clear objective nor identifies a specific gap in the literature. Please begin the Introduction with a concise aim statement and explain what is unknown and why this review is needed now.

Response 1 : Thank you for pointing this out. We agree with this comment. Therefore, we revised manuscript to begin the Abstract with a concise aim statement and explain what is unknown and why this review is needed now. This change can be found Page2, 1st paragraph.

Abstract: Osseointegration is the direct contact between living bone and a dental implant, with supporting evidence confirming the direct connection between bone and titanium found using an electron microscope. However, the fundamental mechanisms and interconnections between the bone and titanium were not clearly understood. At present, osteoimmunology explores the interaction between bone and immune cells not only in the medical field but also in dentistry. Immunology in bone cell formation has long been a research topic; however, interest in these effects has recently surged. As subsequent studies, osteoimmune reaction occurs in response to dental implant insertion into the bone and this mechanism portrays more accurate tissue response compared to the traditional term osseointegration. Additionally, osseointegration is a foreign body defense mechanism to protect the implant when bone forms at the contact surface between the dental implant and the alveolar bone. The term “osteoimmunology” refers to the relationship between the immune system and bone tissues. Understanding osteoimmunologic concepts may enable the development of immunomodulatory strategies to improve, maintain, and ultimately restore osseointegration. In order for biocompatible materials such as dental implants to settle and be maintained in the body, it is necessary to understand the complex interrelationships of the bone immune environment, which will enable the development of biomaterials that are more favorable to osteoimmune environments. Therefore, this review presents previous insights into cellular and molecular interactions between bone and the immune system, specifies the roles of T-regulatory cells (Tregs) and macrophages, and demonstrates their potential for translational applications worldwide.

Comments 2 : The opening paragraph mentions “oste­o­immunology in the field of dentistry,” yet no dental‑specific link is developed. Please make this connection explicit throughout or omit the title of this subsection in the Introduction.

Response 2: Thank you for pointing this out. We agree with this comment. Therefore, we revised manuscript to omit the title of this subsection in the Introduction. This change can be found Page2, 2nd paragraph.

“Osteoimmunology—Fundamentals of Immunity and Immune Cells”

Comments 3 : General immunology of T cells/T regs is reviewed in the first half, but they disappear in the osseointegration section, which shifts to macrophages. Explain why T regs are not discussed in the peri‑implant context and why macrophages are not introduced earlier.

Response 3 : Thank you for pointing this out. We agree with this comment. Just we tried to discuss about Treg and peri-implant context. This can be found Page 4, 3rd paragraph and Page 5, 1st paragraph as like this,

    “ In cases of periodontitis and peri-implantitis, appropriate regulation of immune response prevents the spread of pathogenic microorganisms and simultaneously avoids additional collateral tissue damage. Therefore, Tregs are primarily drawn to the infected tissue to control the immune response [21,22]. Tregs regulate bone metabolism by secreting transforming growth factor-β (TGF-β), interleukin-10 (IL-10), IL-4, and cytotoxic T lymphocyte-associated antigen-4 (CTLA-4) and directly inhibiting osteoclast production. When peri-implantitis progresses, the proportion of Th17 increases. It is known that in the early stages, Tregs maintain a balance with Th17, but in the later stages, as the proportion of Tregs decreases, there is an immune imbalance, leading to bone resorption. Hence, Foxp3+Tregs play a crucial role in regulating periodontitis and peri-implantitis [23,24].”

Espscially on 2.6. Tregs and Macrophages, we tried to describe about Treg and Macropgages containing the connection between Tregs and Macrophagers. This can be found Page 6, 4nd paragraph as like this,

2.6. Tregs and Macrophages Tregs can interact with other major innate immune cells such as macrophages for inflammatory responses. Macrophages have a key role in autophagy, which removes apoptotic neutrophils and other cells. Additionally, they contribute to tissue repair and regeneration [56]. Monocyte-derived macrophages exist in two subsets: M1 and M2. M1 are pro-inflammatory macrophages induced by interferon-γ (IFN-γ) or TNF-α, while M2 are anti-inflammatory macrophages induced by IL-4/IL-13 or IL-10. Here, Tregs are an important regulator for expressing the macrophage phenotypes. They alleviate tissue damage, regulate macrophage activity, and escalate tissue survival to maintain tissue repair and homeostasis [57]. Tregs utilize various types of cytokine secretion or cell interactions to induce monocytes’ phagocytic capacity and markers such as CD206, CD163, and HO-1 while differentiating monocytes into the anti-inflammatory M2 phenotype. This reveals the low expression of CD40, CD80/86, and Class II MHC. Tregs can also increase the secretion of anti-inflammatory cytokines such as IL-4, IL-10, IL-13, and TGF-β and suppress the secretion of pro-inflammatory cytokines like TNF-α, NO, and ROS [58,59]. In the Treg-mediated immune suppression, there exist various cell-contact regulators. For example, CTLA-4 interacts with co-stimulatory receptors CD80/86, PD-1 binds to PD-L1, lymphocyte antigen gene-3 (LAG-3) binds to MHC-II, T cell immunoreceptor with Ig and ITIM domains (TIGIT) binds with CD155/122, FasL induces Fas+ cell apoptosis, and Nrp1 monitors Tregs’ Foxp3 activation and deactivation [60]. However, for the Treg-mediated macrophage inflammation suppression to occur, a signaling pathway via Krüpel-like factor 10 (KLF10) and mammalian target of rapamycin complex 1 (mTORC1) is required [61,62]. Tregs are responsible for directly producing EVs on the cell surface, which leads to the emergence of CD73 in EVs. This triggers the conversion of the extracellular AMP into adenosine and regulates immune responses by inhibiting M1 macrophages and promoting M2 macrophages [63].

Comments 4 : The literature covering immune response and osseointegration is selective and incomplete. The discussion relies almost exclusively on the “foreign‑body response” concept introduced by Albrektsson and colleagues, but provides no evidence or contrasting viewpoints. A large body of work on innate and adaptive immunology during metallic implant osseointegration (both dental and orthopaedic) is omitted. Please broaden the literature survey to include experimental and clinical studies that investigate macrophage and T‑cell dynamics beyond the Albrektsson paradigm; also please integrate findings from orthopaedic models where immune–bone crosstalk is well characterized

Response 4 : Thank you for pointing this out. We agree with this comment. As you commented, we revised the manuscript to provides evidence and innate and adaptive immunology during metallic implant osseointegration (both dental and orthopaedic) including experimental and clinical studies that investigate macrophage and T‑cell dynamics. This can be found Page 9, 2nd and 3rd paragraph as like this,

Clinical and animal studies indicate that many immune cell types, including T cells, macrophages, and neutrophils, are involved in fracture healing by coordinating a chain of events that regulate bone formation and remodeling. Immune cells, including T cells, are involved in fracture healing, regulating the microenvironment and preparing MSCs in the inflammatory phase, and coordinating bone formation and resorption in the bone remodeling phase. (T cell related osteoimmunology in fracture healing-Potential targets for augmenting bone regeneration)

In 2023, Waad Kheder showed that the infiltration of immune cells in the tissues surrounding dental implants plays an important role. There is an increase in lymphocyte and macrophage infiltration in the tissues around the failed implant. In particular, it is associated with T-cell-mediated immunity and M1/M2 macrophage polarization. These results show evidence that titanium particles regulate the polarization of lymphocytes and macrophages in the gingival tissue around the implant, which may help to understand the imbalance in osteoblast and osteoclast activity and the failure of osseointegration of dental implants. (Titanium Particles Modulate Lymphocyte and Macrophage Polarization in Peri-Implant Gingival Tissues)

Comments 5 : The title mentions T regs but not macrophages, although macrophages dominate the second half of the review. Please revise the title.

Response 5 : Thank you for pointing this out. . We agree with this comment. If possible, we want to entitle “Mediation of Osseointegration, Osteoimmunology, and Osteoimmunologic Integration by Tregs and Macrophages: A Narrative Review”, Not only T regs but also Macropgage.

Comments 6 : The abstract does not state explicit review questions or summarise how the manuscript answers them.  Please rewrite to state the aim/gap, to present the main thematic findings, and to conclude with specific knowledge gaps and clinical implications.

Response 6 : Thank you for pointing this out. We agree with this comment. Therefore, we rewrite the abstract to state the aim/gap, to present the main thematic findings, and to conclude with specific knowledge gaps and clinical implications. This change can be found.

Abstract: Osseointegration is the direct contact between living bone and a dental implant, with supporting evidence confirming the direct connection between bone and titanium found using an electron microscope. However, the fundamental mechanisms and interconnections between the bone and titanium were not clearly understood. At present, osteoimmunology explores the interaction between bone and immune cells not only in the medical field but also in dentistry. Immunology in bone cell formation has long been a research topic; however, interest in these effects has recently surged. As subsequent studies, osteoimmune reaction occurs in response to dental implant insertion into the bone and this mechanism portrays more accurate tissue response compared to the traditional term osseointegration. Additionally, osseointegration is a foreign body defense mechanism to protect the implant when bone forms at the contact surface between the dental implant and the alveolar bone. The term “osteoimmunology” refers to the relationship between the immune system and bone tissues. Understanding osteoimmunologic concepts may enable the development of immunomodulatory strategies to improve, maintain, and ultimately restore osseointegration. In order for biocompatible materials such as dental implants to settle and be maintained in the body, it is necessary to understand the complex interrelationships of the bone immune environment, which will enable the development of biomaterials that are more favorable to osteoimmune environments. Therefore, this review presents previous insights into cellular and molecular interactions between bone and the immune system, specifies the roles of T-regulatory cells (Tregs) and macrophages, and demonstrates their potential for translational applications worldwide.

Comments 7 : The review claims that magnesium promotes M2 macrophage polarisation. Recent in‑vivo studies show biodegradable Mg implants elicit a pronounced, transient M1 surge before any M2 shift. Please correct.

Response 7 : Thank you for pointing this out. We agree with this comment. Therefore, we ㅁadded references and corrected the manuscript that recent in‑vivo studies show biodegradable Mg implants elicit a pronounced, transient M1 surge before any M2 shift with references. (92, 93).  This change can be found Page 11, 2nd paragraph as like this,

Recent in‑vivo studies show biodegradable Mg implants elicit a pronounced, transient M1 surge before any M2 shift [92, 93].

Ref. 92

Heithem Ben Amara, et al. Magnesium implant degradation provides immunomodulatory and proangiogenic effects and attenuates peri-implant fibrosis in soft tissues. Bioactive Materials 2023, 26, 353–369

In contrast, iNos/Mrc1 ratio for cells adherent to Mg implants was consistently shifted toward iNos gene between 1 d and 6 d, before decreasing and switching to a pattern similar to that of cells adherent at Ti implants (that is shifted toward Mrc1 gene) at 14 d and 28 d(361p)

Ref. 93

Maryam Rahmati, et al. Early osteoimmunomodulatory effects of magnesium–calcium–zinc alloys. Journal of Tissue Engineering 2021, Volume 12: 1–19.

We studied the macrophage polarization in both groups by focusing on the distribution of macrophage type 1–2 subsets over time (Figures 1(a) and S1-A). We used CD68 and CD80 subsets as the biological markers of type 1 macrophages, as well as CD68 and CD206 subsets for type 2 macrophages.41 During the first 5 and 10 days, macrophage phenotype changed in both groups from predominantly macrophages type 1–2 with significant changes for Mg group (Figure 1(a)–(d)). We observed a significantly higher number of CD206-positive macrophages (type 2 macrophages) for Mg group in comparison to sham over time. Type 2 macrophages were present 2 and 5 days after the surgery, and they dominated at the interface 10days after surgery, while the percentage of type 1 macrophages was decreased significantly. Although the same pattern was seen in the sham group, the changes in type 2 macrophages were not significant indicating improved immunomodulatory effects of Mg alloy toward tissue healing. (5p)

Reviewer 2 Report

Comments and Suggestions for Authors The article entitled: “Mediation of Osseointegration, Osteoimmunology, and Osteo immunologic Integration by Tregs and Macrophages” has the potential to be a valuable contribution to the literature. The manuscript was beautifully written, but I have a few suggestions: 1. The Abstract is a little bit constrained (97 words). I suggest extending it to close to 250 words. 2. In addition, I suggest removing author citations (Arron and Choi in 2000) in the Abstract. 3. As far as only a selected group of articles were mentioned to support authors' perspectives, I suggest stating (usually in the title) that the manuscript is a narrative review. Otherwise, the manuscript was well written, and the English language is perfectly clear as well. The text has an adequate length, it is well organized, and most of all, it is perfectly clear (I would say: didactic) in a research field usually dominated by letters soup (ILs, GFs,…) and cell biology techniques. So, it has the potential to be really accessible to the clinicians.

Author Response

Thank you for your review. It was great honor for us to meet your comments.

Comments and Suggestions for Authors

The article entitled: “Mediation of Osseointegration, Osteoimmunology, and Osteo immunologic Integration by Tregs and Macrophages” has the potential to be a valuable contribution to the literature. The manuscript was beautifully written, but I have a few suggestions:

Comments 1 : The Abstract is a little bit constrained (97 words). I suggest extending it to close to 250 words.

Response 1 : Thank you for pointing this out. We agree with this comment. Therefore, we revised The Abstract extending it to close to 250 words. This change can be found

Abstract: Osseointegration is the direct contact between living bone and a dental implant, with supporting evidence confirming the direct connection between bone and titanium found using an electron microscope. However, the fundamental mechanisms and interconnections between the bone and titanium were not clearly understood. At present, osteoimmunology explores the interaction between bone and immune cells not only in the medical field but also in dentistry. Immunology in bone cell formation has long been a research topic; however, interest in these effects has recently surged. As subsequent studies, osteoimmune reaction occurs in response to dental implant insertion into the bone and this mechanism portrays more accurate tissue response compared to the traditional term osseointegration. Additionally, osseointegration is a foreign body defense mechanism to protect the implant when bone forms at the contact surface between the dental implant and the alveolar bone. The term “osteoimmunology” refers to the relationship between the immune system and bone tissues. Understanding osteoimmunologic concepts may enable the development of immunomodulatory strategies to improve, maintain, and ultimately restore osseointegration. In order for biocompatible materials such as dental implants to settle and be maintained in the body, it is necessary to understand the complex interrelationships of the bone immune environment, which will enable the development of biomaterials that are more favorable to osteoimmune environments. Therefore, this review presents previous insights into cellular and molecular interactions between bone and the immune system, specifies the roles of T-regulatory cells (Tregs) and macrophages, and demonstrates their potential for translational applications worldwide.

Comments 2. In addition, I suggest removing author citations (Arron and Choi in 2000) in the Abstract.

Response 2 : Thank you for pointing this out. We agree with this comment. Therefore, we removed author citations (Arron and Choi in 2000) in the Abstract.

Comments 3 : As far as only a selected group of articles were mentioned to support authors' perspectives, I suggest stating (usually in the title) that the manuscript is a narrative review.

Response 3 : Thank you for pointing this out. We agree with this comment. We stated that the manuscript is a narrative review in the title as you suggested.

“Mediation of Osseointegration, Osteoimmunology, and Osteo immunologic Integration by Tregs and Macrophages: A Narrative Review” 

Otherwise, the manuscript was well written, and the English language is perfectly clear as well. The text has an adequate length, it is well organized, and most of all, it is perfectly clear (I would say: didactic) in a research field usually dominated by letters soup (ILs, GFs,…) and cell biology techniques. So, it has the potential to be really accessible to the clinicians.
